# Quantifying the Health Burden Misclassification from the Use of Different PM_2.5_ Exposure Tier Models: A Case Study of London

**DOI:** 10.3390/ijerph17031099

**Published:** 2020-02-09

**Authors:** Vasilis Kazakos, Zhiwen Luo, Ian Ewart

**Affiliations:** School of Built Environment, University of Reading, Reading RG6 6DF, UK; v.kazakos@pgr.reading.ac.uk (V.K.); i.j.ewart@reading.ac.uk (I.E.)

**Keywords:** PM_2.5_, population exposure, tier-models, health burden misclassification, BenMap-CE

## Abstract

Exposure to PM_2.5_ has been associated with increased mortality in urban areas. Hence, reducing the uncertainty in human exposure assessments is essential for more accurate health burden estimates. Here, we quantified the misclassification that occurred when using different exposure approaches to predict the mortality burden of a population using London as a case study. We developed a framework for quantifying the misclassification of the total mortality burden attributable to exposure to fine particulate matter (PM_2.5_) in four major microenvironments (MEs) (dwellings, aboveground transportation, London Underground (LU) and outdoors) in the Greater London Area (GLA), in 2017. We demonstrated that differences exist between five different exposure Tier-models with incrementally increasing complexity, moving from static to more dynamic approaches. BenMap-CE, the open source software developed by the U.S. Environmental Protection Agency, was used as a tool to achieve spatial distribution of the ambient concentration by interpolating the monitoring data to the unmonitored areas and ultimately estimating the change in mortality on a fine resolution. Indoor exposure to PM_2.5_ is the largest contributor to total population exposure concentration, accounting for 83% of total predicted population exposure, followed by the London Underground, which contributes approximately 15%, despite the average time spent there by Londoners being only 0.4%. After incorporating housing stock and time-activity data, moving from static to most dynamic metric, Inner London showed the highest reduction in exposure concentration (i.e., approximately 37%) and as a result the largest change in mortality (i.e., health burden/mortality misclassification) was observed in central GLA. Overall, our findings showed that using outdoor concentration as a surrogate for total population exposure but ignoring different exposure concentration that occur indoors and time spent in transit, led to a misclassification of 1174–1541 mean predicted mortalities in GLA. We generally confirm that increasing the complexity and incorporating important microenvironments, such as the highly polluted LU, could significantly reduce the misclassification of health burden assessments.

## 1. Introduction

There is growing evidence that air pollution and specifically fine particulate matter (PM_2.5_) contribute significantly to health burden and further, there is a close relationship between long-term air pollution exposure and adverse health effects in urban populations [1,2]. The assessment of Global Burden of Disease (GDB) indicated that PM_2.5_ contributed 4.24 million deaths globally in 2015 [3]. Assessments of human health effects attributed to an air pollutant are dependent on the magnitude of human exposure to that pollutant. Thus, the accuracy of a health burden assessment is determined by the uncertainty of predicted population exposure. Quantifying the population exposure to air pollution is subject to several challenges:

The spatiotemporal variability of ambient concentration is strongly influenced by emissions dynamics, predominantly from road transport, (such as peaks in traffic-related pollution during rush hours), meteorological conditions, which determine the transport and dilution of air pollutants and local conditions such as the urban form (e.g., the presence of high buildings can reduce the dispersion of the pollutants), which are the most important factors leading to significant variation of air pollutants in urban areas.

The proportion of outdoor air infiltrated to indoor microenvironments (MEs) is influenced by different housing designs and patterns of behaviour inside the building. 

The spatiotemporal variability of people’s activity (population time–activity patterns) in various MEs [4].

Around 75% of European populations live in cities, with a highly variable range of activities carried out at different times and in different places [5]. The quality of data, or the absence of key components within an epidemiological exposure assessment, is likely to affect the magnitude and significance of the prediction misclassification in a health burden assessment (Figure 1).

Traditionally, epidemiological studies relied on centralized ambient concentration measurements of limited monitoring sites [6,7,8,9,10]. This is likely to lead to an exposure error, since several monitoring studies have suggested that air pollution data from a single site can represent only a small surrounding area especially in urban environments, due to pollutants’ spatial heterogeneity [11,12]. Ambient air pollutant concentration can be estimated in several ways such as through field observations, statistical modelling such as land-use regression (LUR) and air quality dispersion models (AQM) that can use various spatial resolutions [13]. Willers et al. [14] indicated that using air quality data measured at a single site and assuming that exposure across cities was the same, could cause considerable misclassification of exposure. In their study, they examined the difference in mortality risk between neighborhoods in the city of Rotterdam and found that the mortality risks between neighborhoods had a difference of up to 7%. By utilizing land use regression techniques and air quality models, several studies have managed to demonstrate that an increased spatial resolution of the exposure concentration could lead to significantly different exposure or health burden estimates [15,16,17,18]. Similarly, Punger and West [19] assessed the effect of spatial resolution to population-weighted PM_2.5_ concentrations in the U.S. by utilizing the Community Multiscale Air Quality (CMAQ) model. They found that population exposures, maximum concentrations and standard deviations all reduced at coarser resolutions. At 408 km resolution, exposure and maximum concentration were 27% and 71% lower, respectively than those at 12km resolution. Attributable mortality also reduced as the resolution became coarser. Several studies have shown that coarse resolutions might result in lower mortality attributed to PM_2.5_ [20]. Fenech et al. [21] concluded that total mortality estimates were sensitive to model resolution up to ±5% across Europe, whereas Korhonen et al. [22] found that, considering only local sources of primary PM_2.5_, the mortality reduced by 70% in the whole country (Finland) and 74% in urban areas when the resolution changed from 250 m to 50 km.

Apart from the exposure misclassification due to the different levels of spatiotemporal resolution of outdoor concentration, there are other significant contributors, in particular the infiltration of outdoor pollutants to indoor MEs and different time-activity patterns in MEs. As particles infiltrate and persist indoors, where people living in urban areas spent over 80% of their daily time [23], most of the exposure to PM_2.5_ actually occurred in the indoor microenvironments [23,24,25]. The fraction of ambient PM_2.5_ that infiltrates indoor microenvironments can vary due to particle size, building characteristics, meteorological conditions and human activities [26]. Consequently, relying on outdoor measurements alone can therefore lead to exposure misclassification. Moreover, variations in the time spent in various MEs (e.g., outdoors, indoors, vehicles, subway) also influence population exposure to outdoor-generated PM_2.5_ due to the spatial variability of both outdoor concentrations and the indoor transport of ambient PM_2.5_. Baxter et al. [27] compared four different approaches to PM_2.5_ exposure prediction, where each model was of a different complexity. In their study they focused on the heterogeneity in exposures but did not investigate the influence on health effect predictions. They suggested that geographic heterogeneity in both housing stock (and thus a relatively consistent Air Change Rate) and human activity patterns contribute to significant heterogeneity in ambient PM_2.5_ exposure both within and between cities that is not demonstrated by stationary monitors. Ma et al. [28] compared three different types of PM_2.5_ exposure estimates to illustrate the differences in exposure levels between estimates obtained from different approaches. They found that the daily average PM_2.5_ exposures for residents with different activity patterns may vary significantly even when they were living in the same neighborhood. Several studies have also investigated the correlation between outdoor PM_2.5_ and mortality, although their results are skewed by the fact that people spend the majority of the time indoors. Ji and Zhao [29] used existing epidemiological data on ambient PM_2.5_-related mortality to estimate mortality associated with indoor exposure to outdoor-generated PM. This was the first attempt to quantify that relationship and their results indicated that outdoor PM had substantial effects on health caused by exposure within indoor MEs. Recently, Fenech and Aquilina [30] used the annual mean PM_2.5_ concentrations derived from local fixed monitoring stations to estimate the PM_2.5_-related mortality in the Maltese Islands. They found that the attributable fraction of all-cause mortality associated with long-term PM_2.5_ exposure ranged from 5.9% to 11.8%, indicating that PM_2.5_ concentration is a major component of attributable deaths. Azimi and Stephens [31] used a modified version of the common exposure-response function and developed a framework for estimating the total U.S. mortality burden attributed to exposure to PM_2.5_ of both indoor and outdoor origins. They found that residential exposure to outdoor-generated PM_2.5_ accounted for 36% to 48% of total exposure, indicating that efforts to mitigate mortality associated with exposure to PM_2.5_ should consider indoor pollution control as well.

That of particular importance is how different exposure approaches impact long-term health burden/mortality predictions and the magnitude of the resultant impact. We made multiple comparisons between refined ambient PM_2.5_ exposure surrogates (that account for important factors such as the infiltration and time-activity) and the fixed-site monitor PM_2.5_ concentrations to indicate the importance of including more dynamic data to epidemiological studies and to demonstrate how more complex modelling approaches modify mortality predictions. By using BenMap-CE we were able to provide the spatial distribution of health outcomes influenced by the exposure misclassification. While a number of studies have already investigated exposure misclassification when using different approaches and others have estimated health effects based on specific exposure metrics, the aim of this work is to move one step further and answer the question: how much is the misclassification that occurs when using different exposure approaches to predict health burden?

## 2. Materials and Methods 

This work aims to quantify the long-term health burden misclassification that occurs when different PM_2.5_ exposure metrics are utilized. An ecologic design was used to generate associations between air pollution exposure and health outcomes. We investigated the Greater London Area (GLA), building on recent exposure studies that have explicitly estimated London population exposure using hybrid dynamic models [32]. Here, we have described five different exposure Tier-models of incrementally increased complexity are considered by gradually including data of important MEs, such as infiltration rates of the different dwelling types and the London Underground, where London’s population spend most or part of their daily time. The London Travel Demand Survey (LTDS) space-activity data were categorized into three major ME groups. The analysis estimated the magnitude of the change (i.e., avoided or incurred) in mortalities when moving from the central-site monitored concentrations as a surrogate for population exposure (Tier-model 1) to more refined exposure Tier-models. The original ambient PM_2.5_ concentrations were based on average hourly data measured by 23 monitoring stations located in the GLA [33] and the examined MEs were: i) indoors (i.e., home-indoor), ii) aboveground transportation iii) the London Underground and iv) outdoors. The following sections describe the structure of the methodology and the development of each component.

### 2.1. Developing Tier Models to Estimate Human Exposure

To capture different exposure assessment methods that have been used in epidemiology, we developed five different Tier models of increased complexity, moving from static to more dynamic approaches (Table 1). This method was separated into two parts: i) The microenvironments and time-activity patterns were classified and calculated based on the derived information; ii) the time-activity information was matched with corresponding microenvironmental concentrations to estimate the dynamic time-weighted exposure. The exposure time was considered costly and the metrics estimated the annual hourly-average PM_2.5_ exposures, which were then used as an input for BenMap-CE [34].

The Tier-model stages and the respective approaches are briefly described below.

Tier model 1: Outdoor
E = C_out_,(1)
where E is mean exposure and C_out_ is mean outdoor concentration of PM_2.5_.

Hourly readings were extracted from the London Air Quality Network (LAQN) [33]. LAQN consists of automatic monitoring equipment in fixed cabins, which measures air pollution at breathing height. It provides electronically available data on concentrations of major urban pollutants and has been used in several studies [35,36]. The ratified concentration data from 23 available monitoring stations in GLA were downloaded and added to BenMap-CE. Only the monitors that could provide at least 70% of the data for the whole year were selected. The ambient concentration was considered as representative of the total population exposure.

Tier model 2: Indoor
E = C_in_,(2)
where C_in_ is the mean indoor (i.e., home-indoor) concentration.

This Tier model utilized the information of the spatially distributed concentration and the total average Indoor/Outdoor (I/O) ratio in GLA to estimate the exposure inside the residence [37].

Tier model 3: Indoor (dwellings)
E = ∑C_out_* F_i_* x_i_,(3)
where F_i_ is the infiltration rate of each dwelling type (i) and x_i_ is the frequency (%) of this type in London.

In this study, all the indoor environments were combined into one single ME (i.e., home-indoor) without considering other indoor environments, such as office or commercial buildings, due to the lack of infiltration data. Subsequently, the I/O ratios that we used also represented offices and other indoor places, assuming that the I/O ratios for other indoor MEs had the same values as domestic home buildings [32]. The I/O ratios of London’s housing stock were obtained from Taylor et al. [37]. In their study they estimated the Indoor/Outdoor ratio of 15 building archetypes. We grouped these archetypes into five main dwelling types in response to available housing stock data in Middle-Super-Output-Area resolution obtained from the Mayor of London, Datastore [38]: i) flat, ii) bungalow, iii) terraced, iv) semi-detached and v) detached (Table 2). The frequency of each type could be calculated from the number of properties in the GLA, which represented 98.7% of the housing (The average I/O ratio was assigned to the unknown 1.13%). Figure 2 shows the annual average I/O ratios of PM_2.5_ concentration in the GLA. The average ratios, including all dwelling types and their frequency, ranged from less than 0.54 to 0.59. The highest ratios were observed in Outer London, whereas the lowest ratios were observed in Inner and South West London, probably due to the newer building stock and the large number of flats in large buildings (London Datastore), where the available surface for infiltration was considerably smaller.

Tier model 4: Outdoor + Indoor + Transportation (aboveground and underground)
E = (C_out_* t_out_) + (∑C_out_*F_i_ *x_i_) * t_ind_ + (∑C_out_* F_j_) * t_abg_ + (C_undg_* t_undg_),(4)
where (j) is each aboveground transport-ME (tMEs) and t_out_, t_ind_, t_abg_ and t_undg_ is the fractional time spent (%) annually outdoors, indoors, aboveground tME and London Underground (LU) tME, respectively.

This Tier-model includes transportation as an additional microenvironment, where an urban population spends time during the day. This ME was categorized into aboveground and underground transportation. Aboveground transportation refers to car, bus and train, whereas underground to London subway. By separating transportation into 2 groups we were able to evaluate the influence of a highly polluted ME, like the London Underground (described in the next section), on the total population exposure concentration.

The space–time–activity data for our study were based on the London Travel Transport Agency (LTDS) of Transport for London (TfL) [39] for the period between 2005 and 2010 (Table 3). The data were generated from the interviews of approximately 8000 households per year, providing very useful information about their daily time–activity patterns, including travel modes and trip times. The data were scaled to represent the population of London, excluding children under five years old [32].

According to Smith et al. [32], the average daily percentage of time spent indoors was 95.7 %, whereas people spent 2.5%, 0.4% and 1.4% in aboveground transportation, London Underground and outside (walking or cycling), respectively. This proportion of time spent indoors also includes approximately 20% of surveyed people, who did not leave their house. In this study, these percentages were used as annual averages for the whole population over five years old, including the different times spent during weekdays and weekends.

For the in-vehicle exposure of the aboveground sub-microenvironment, we calculated the PM_2.5_ concentration by solving the mass balance equation [30]: dC_in_ / dt = λ_win_ * (C_out_ − C_in_) − ηλ_HVAC_ * C_in_ − V_g_ * (A’ / V) * C_in_ + Q/V,(5)
where C_out_ is the outdoor concentration around the vehicle, C_in_ the concentration inside the vehicle, λ_win_ and λ_HVAC_ are the hourly air exchange rates from the windows and mechanical ventilation system, respectively, n is the filter removal efficiency taking values between 0–1, V_g_ is the deposition velocity in (m/h), A’ is the internal surface area, V is the volume of the vehicle and Q is the in-vehicle particle emission rate in μg/h. To solve this equation, the same values with Smith et al. [32] were used except for the concentrations and the commuter’s surface was derived from Song et al. [40], in order to calculate A’.

Tier model 5: Outdoor + Indoor + Transportation (aboveground and underground→ deep lines + subsurface lines).

The time-weighted exposure equation associated with this Tier model stage is:E = (C_out_* t_out_) + [(∑C_out_* F_i_ * x_i_) * t_ind_] + (∑C_out_* F_j_) * t_abg_ + (C_undg-hvac_ * t_undg-hvac_) + (C_deep-undg_ * t_deep-undg_),(6)

In the 5th and most complex Tier model, the same procedure as in Tier 4 was followed but the London underground microenvironment was further divided into subsurface and deep lines to reflect the significant difference in concentration on two types of lines. The use of mechanical ventilation in the subsurface lines results in much lower PM_2.5_ concentrations than the deep lines due to air filtration (explicitly described in the next section). Hence, by dividing the underground into two subgroups we were able to improve the exposure estimates and to examine the contribution of a very highly polluted microenvironment to the total exposure. The proportion of time spent in each of those two subcategories was assumed according to the number of annual journeys completed in each line during 2017, where 77% were made by the deep-line underground and 33% by the subsurface.

#### PM_2.5_ Concentration in the London Underground

As the London Underground microenvironment was unable to be accurately represented by the outdoor measurements, due to its high concentration of PM_2.5_ and its limited connection to the outside world, a series of air pollution measurements were conducted inside the London Underground. The PM_2.5_ measurements took place on five major London Underground platforms and trains (Bakerloo line, Circle line, Central line, District line and Victoria line) by using the portable DustTrak II Aerosol Monitor 8534, a light scatter laser photometer, which could provide a large number of real-time readings. The current selection of the lines was decided in order for both the deep without mechanical ventilation lines and the subsurface with HVAC lines to be represented by our measurements. 

Our original intention was that the measurements would reflect the cold and the warm period of 2017. Hence, the experiment was conducted during the morning and the afternoon for one week in February and one week in July. The average concentration in the London Underground for the whole year was very high, approximately 218 μg/m^3^, albeit when we grouped the lines into deep without HVAC lines (Central, Bakerloo and Victoria) and subsurface lines with HVAC (Circle, District) we noticed a remarkable difference between the two concentrations (70.2 μg/m^3^ for the subsurface lines and 365.6 μg/m^3^ for the deep lines). The PM_2.5_ concentration levels in the unmeasured lines were assumed to be similar to these measured. The classification of the unmeasured lines was made according to their depth and ventilation system. 

In the London Underground, Seaton et al. [41] reported higher platform concentrations of 480 μg/m^3^. Recently, Smith et al. [42] assessed day to day variation in LU concentrations and compared them with those above ground. During their campaign, 22 repeat journeys were made on weekday mornings over a period of five months. They found that the subsurface ventilated District line had the lowest PM_2.5_ concentration levels (i.e., mean 32 μg/m^3^) and the deep unventilated Victoria line the highest (i.e., mean 381 μg/m^3^), while the mean concentration in the LU, according to their measurements, was 302 μg/m^3^. Although their monitoring method and equipment were different from those used in this study and the sampling period was longer, their findings do not differ significantly from ours. Even though the station measurements in the UK are limited, most of the studies made so far have measured approximately two times higher concentrations in the London Underground than in other undergrounds worldwide [43,44], probably due to its age and the limited ventilation systems.

### 2.2. Simulating PM_2.5_ Exposure Concentration and Estimating Health Impact Using BenMap-CE

The environmental Benefits Mapping and Analysis Program—Community Edition (BenMap-CE) is a powerful Geographical Information system (GIS)-based program that estimates the health effects associated with the change in air quality [34,45]. These data consisted of a middle layer super output areas (MSOA) map of GLA, the derived monitoring data and London’s population data, in order to estimate the health impact. BenMap-CE provides three interpolation methods: the closest monitor, the fixed radius, and Voronoi Neighbour Averaging (VNA). Among the incorporated methods, VNA was the most suitable for our case, covering the unmonitored areas and giving the best spatial distribution of the concentration. 

After uploading the essential data and determining the appropriate Health Impact Function (HIF) for our analysis, we were able to quantify the health impact misclassification (i.e., change in all-cause mortality, either incurred or avoided) resulting from the exposure metric differences. In this study, the following long-term health impact function was used to estimate the change in all-cause mortality [46]:ΔY = Y_0_* (1 − *e*^−^^*β*Δ^^*PM*^) * Pop,(7)
where ΔY is the change in health effect, Y_0_ is the baseline mortality rate (the mortality rate at minimum risk concentration), *β* is the unitless beta coefficient, *ΔPM* is the change in the exposure rates between Tier 1 and the other Tier models (Tier 1 is the base case) and Pop is the exposed population. 

One limitation of the aforementioned effort to estimate the health impact of indoor air pollution is the use of the mortality effect estimate (i.e., beta coefficient) that is usually taken directly from the epidemiology literature on the studies conducted for outdoor air pollution. Therefore, to account for that fact, some studies on the health effects of outdoor-generated PM_2.5_ introduced a method for modifying the mortality effect estimate (i.e., beta coefficient) based on the average infiltration factor combined with the mean fraction of time spent in indoor MEs [13,47,48]. However, the application of the adjusted coefficient is solely for the component of indoor PM_2.5_ of outdoor origin and not of indoor PM_2.5_ in total. The way indoor particle sources are treated has a larger impact than the adjustment of the coefficient for the outdoor-generated fine particles and remains an evidence gap of considerable public health importance. In another study, Logue et al. [49] used a central estimate of the beta coefficient for premature mortality related to both indoor- and outdoor-generated PM_2.5_, which was directly derived from the epidemiology literature. In our case, due to the mobile monitoring conducted in the LU and the distinct function of BenMap-CE, a central mortality effect coefficient from Pope et al. [50] was used as an input. The mortality effect coefficient was utilized to generate BenMap’s health impact functions in the direction of estimating the change in estimates of mortality (either avoided or incurred) when using different exposure metrics. Furthermore, we estimated the percentage decrease in the predicted avoided cases when moving from the less complex (static) metrics to more dynamic metrics.

## 3. Results

### 3.1. Exposure Metrics Summary

The highest annual average exposure concentration was approximately 13.1 μg/m^3^ for Tier model 1. Tier model 2 and Tier model 3 indicated that the exposure that occurred indoors was much lower than outdoors due to the infiltration rates of the buildings, resulting in annual average exposure concentrations of 7.18 μg/m^3^ and 7.26 μg/m^3^, respectively. There was an approximately 45% reduction between Tier 1 and Tier 3. This result clearly suggests that spending long periods of time indoors, reduces the exposure to outdoor-generated air pollution. The incorporation of transportation and predominately the highly polluted London Underground in Tier model 4 resulted in an elevated exposure concentration (8.28 μg/m^3^), pinpointing that even though the time spent in transit is only 2.9%, this microenvironment has a significant contribution to the total exposure. By dividing the London Underground into subsurface with HVAC and deep line without HVAC, we were able to quantify the impact of the most highly polluted ME on the total exposures (the deep-line underground). Tier 5 showed an approximately 0.30 μg/m^3^ higher exposure concentration (8.60 μg/m^3^) than Tier 4, where an average concentration for the whole underground was used (Table 4).

PM_2.5_ exposure concentration maps for each Tier-model stage were created by BenMap-CE showing how the exposure was distributed across GLA. Figure 3a,b illustrate the spatial distribution of the annual exposures in Tier 1 and 5. The maps of Tiers 2, 3 and 4 are included in the Supplementary Information (Figure A1a–c in the Section A.1). The highest exposure concentrations occurred in Inner London for both Tier 1 and Tier 5 (15.4 μg/m^3^ and 10.1 μg/m^3^, respectively), whereas the lowest exposures were observed in Western GLA (less than 10.9 and less than 7.10 μg/m^3^ for Tier 1 and 5, respectively). The incorporation of indoor infiltration along with time-activity data led to an overall mitigation of the exposure concentrations in GLA when Tiers 2, 3, 4 and 5 were used. After the utilization of our most complex model, Tier 5 had the highest difference observed at the centre with approximately 37% (Figure 3c), while average reduction in GLA was approximately 34%. Inner London continued to show the highest values (Figure 3b), although the infiltration factors in Inner London were lower than in the outskirts. This could be due to the much higher outdoor concentrations in Inner GLA than in the Outer. In Inner London, the higher number of sources of anthropogenic and traffic-related pollutants, including PM_2.5,_ generate significantly higher ambient pollution levels. Several studies suggest that traffic pollutants are elevated above background concentrations around major roads and highways [13,51]. The percentage of exposure concentration reduction in Tiers 2, 3 and 4 after comparison with our baseline exposure concentration (Tier 1), is illustrated in Figure A2a–c in the Section A.2. Apart from proximity to roads, fewer green spaces and the densely constructed city center may also contribute to the higher levels of outdoor particulate pollution [52,53,54]. Urban populations are subject to daily activity patterns, so that exposure is not a static phenomenon but should be quantified as a function of concentration and time [4]. Therefore, by assigning people’s exposure to a single location (e.g., at their residence) and ignoring highly polluted MEs such as the subway, it is unlikely to accurately represent total exposure. Hence, by gradually incorporating time-activity data and indoor MEs, the spatial variability of the exposure concentration across GLA increased. Since we used annual average time-activity data for the entire GLA, time-activity could not change the spatial pattern of the exposure. In our case, the spatial variability of the housing stock and I/O ratios across GLA were the main reasons for any increase in the spatial variability of the exposure concentration.

Figure 4 presents the contribution of each examined microenvironment to the total exposure estimated by Tier-model 5. Indoor exposure concentration is clearly the dominating contributor (approximately 83%) to the total exposure (due to the time that people spent there–95.7%) followed by the deep-line underground ME (14%) albeit people spent on average only 0.31% of their annual time. According to our measurements, the PM_2.5_ concentration in deep underground lines was around 28 times higher than the outdoor levels, which rationalized the high contribution of that ME to total exposure. In contrast, London population spent only 1.4% of its annual time outside and the outdoor ME contributed only 2% to the total exposure concentration. The findings described above indicate that outdoor PM_2.5_ levels are unlikely to accurately represent the total exposure of an urban population like in London.

### 3.2. Epidemiological Implications and Health Impact Misclassification

Because in epidemiology the concentration from central-site monitors is used as a proxy for the exposure to air pollution, we selected Tier 1 as our reference and compared it with the estimates of Tiers 2, 3, 4 and 5. The mean change in the estimates of all-cause mortality when applying Tier model 2 was predicted to be 1541 (95% CI: (427–2633)) deaths, while when using Tier 3 exposure concentration estimates the death cases were reduced to 1521 (95% CI: (421– 2598)). The impact on mortality when applying the 4^th^ Tier model, which included the transportation microenvironments (tMEs), was estimated to be 1257 (95% CI: (347–2151)) cases. Due to the significance of the deep-line underground, the most complex Tier model 5 presented the lowest number of cases compared with the other 3 metrics (Tiers 2, 3 and 4). Namely, once Tier 5 was applied the prediction for the estimated avoided mortalities were 1174 (95% CI: (324 – 2010)). We can assume that the calculated change in mortality represents the potential health burden misclassification that might occur when changing the exposure metrics to assess the population exposure. Subsequently, we were able to estimate the percentage decrease in predicted mortalities when altering the exposure metric’s complexity. The substantial changes in avoided mortality predictions indicate that using a static exposure approach in a study might lead to significant uncertainty in a health burden assessment. As anticipated, the predicted mortality was significantly reduced when increasing the model complexity. The highest changes were observed in Tier-model 2 and 3, due to the time that people spent indoors in urban areas, the big difference between outdoor and indoor exposure and the absence of highly polluted transportation MEs, pinpointing the importance of taking into serious consideration the exposure that occurs inside buildings when estimating health effects. The model predicted most avoided cases when Tier 2 was applied and while increasing complexity the cases showed a decrease of 1.95%, 18.4% and 23.8% for Tier 3, 4 and 5, respectively. As explained above, the London Underground contributes significantly to the total average exposure concentration of the study population by increasing the estimates. Therefore, we can securely presume that this is the main reason for the high decrease in avoided mortalities when Tier 4 and, predominantly, Tier 5 were used in BenMap-CE. 

All results are summarized in Table 5.

Looking at the spatial distribution of the predicted change in mortalities shown in Figure 5 (Tier 1–Tier 5) we can notice that the biggest change in mortality occurred in central GLA. Several factors could explain this result such as the outdoor PM_2.5_ concentration, the housing stock (I/O ratios) and the population. As described above, after the inclusion of the time-activity data there was an overall reduction in exposure concentration because people usually spend most of their time (>95%) in indoor MEs (excluding transportation), where the concentration of outdoor PM_2.5_ is lower than the measured ambient levels. Because the health impact function used by BenMap-CE is a concentration response function, the amount of the reduced exposure concentration determines the fraction of the mortality reduction. In our case, knowing that moving from Tier 1 to Tier 5 would result in a greater reduction of exposure concentration that appeared in central GLA (Figure 3c), we could presume that the high outdoor PM_2.5_ concentration and the building type of that area, were largely responsible for the mortality change. The similar distribution patterns between Figure 3c and Figure 5 also supported this argument. As already shown in Figure 2, the infiltration factors of the buildings there were lower than the rest of the GLA, leading to higher mitigation of the exposure concentration. In the Section A.3, Figure A3a–c show the spatial distribution of the predicted change in mortality between Tier 1 and Tiers 2, 3 and 4, respectively.

Overall, these outcomes demonstrate the importance of the complexity of an exposure metric when incorporated into an epidemiological study. Here, we proved that indoor MEs such as the home and the subway are governing human exposure to air pollution and any possible absence in a metric is likely to cause considerable misclassification of the magnitude of mortality.

## 4. Discussion

Due to the limited time most people spend outside, the amount of ambient concentration of PM_2.5_ that people are directly exposed to is likely to be different based on variation in people’s behavior and the performance characteristics of the buildings they are occupying [55]. Consequently, spatial variability, time-activity and losses due to outdoor-to-indoor transport are all sources of exposure uncertainty in the epidemiological analysis, when fixed-site monitor concentrations are used as surrogates for exposure to air pollution. In this work we established a more comprehensive understanding of population exposure concentration and the impact that different exposure metrics can make on all-cause mortality predictions. We showed that the I/O ratios and individual’s patterns of movement play a key role in estimating exposure to PM_2.5_ and that transportation-MEs, predominately the highly polluted London Underground, are important in accurately establishing exposure. We demonstrated that subway and Indoor MEs make a significant contribution to the exposure misclassification and therefore mortality change predictions. Azimi and Stephens [31] highlighted the importance of including indoor MEs when estimating the total exposure and the need for a better understanding of how the infiltration factors vary by building type in order to improve the exposure estimates and reduce the uncertainty. Based on field measurements, they found that exposure to PM_2.5_ of outdoor origin inside the residence contributed around 67% to the total U.S. mortality burden. In our analysis, we found that the Indoor environment contributed approximately 83% to the total mortality burden in London. The difference in our results may be explained by the different MEs considered in each study. As our aim was to quantify the misclassification and give an insight into how the absence of significant MEs from an exposure assessment could increase the uncertainty, we mainly focused on the different infiltration factors of home types and the LU. Martins et al. [56] determined the PM_2.5_ exposure and estimated the daily PM_2.5_ dose during Barcelona subway commuting. They estimated that the PM_2.5_ dose received by an adult in the subway contributed approximately 46% to the total daily dose in the respiratory tract. In our study, LU contributed approximately 15% to the total health burden. Due to the different methods used and different health endpoints, their results cannot directly be compared to ours. However, their outcomes indicate the non-trivial contribution from subway ME on health effects estimates. Several studies have compared static (home address-based) with more dynamic air pollution methods and proved that there is a reduction in average total exposure levels in urban areas with related characteristics as GLA [32,57]. Tang et al. [57] used a staged modelling approach to evaluate the use of static ambient concentrations as exposure estimates and examined the impact of dynamic components on estimated air pollution exposure. They found that the mean population exposures in Hong Kong for their full dynamic model were approximately 20% lower than the ambient baseline estimates of the static approach. Smith et al. [32] combined a dispersion modelling approach with building infiltration factors and travel behavior in order to create the London Hybrid Exposure Model (LHEM). They found that their model’s estimates were around 37% lower for PM_2.5_ than the static approach (residential address-based). Similarly, by adopting a staged modelling approach to evaluate the effect of including dynamic components to our exposure models we found that the absence of mobility and infiltration factors in the static Tier-model 1 led to an overestimation of annual PM_2.5_ population exposure. Overall, the exposure estimates of our most complex model (Tier 5) were around 34% lower than those of the static baseline model (Tier 1). These findings were different from Tang et al.’s [57] study but very similar to the LHEM study, mainly because the study population was the same and similar travel behavior data was used. Recently, Singh et al. [58] quantified the population exposure to PM_2.5_ concentrations in London and assessed the importance of including movement and indoor infiltration to total population exposure. They found that their refined exposure assessment predicted 28% lower total population exposure than the traditional static exposure method. As in this study, the time-activity data were derived from the LTDS [39] and the study area was London. However, the small difference between their results and ours could be explained by the different datasets used for the infiltration factors and the different concept used for the key MEs (e.g., the London Underground). Results from other similar studies are difficult to find as we compare different exposure estimates during the same time period (2017) in an effort to examine the effect on all-cause mortality predictions. We showed that using a static exposure metric instead of a more dynamic approach (based on time-activity data and indoor infiltration) to predict the mortality in the GLA population would lead to an overestimation of 1174–1541 mean predicted estimates of mortality attributed to PM_2.5_. Ebelt et al. [59] found for several health outcomes associated with cardiopulmonary diseases, analyses with ambient exposures resulted in larger effect estimates. These results strongly supported their original hypothesis that the reduced exposure misclassification resulting from the utilization of ambient exposures instead of ambient concentrations provide more precise estimates of effects in epidemiology.

This work provides further understanding as to the impact of an exposure assessment on the mortality predictions and helps to mitigate the uncertainty in health risk assessments of air pollution. As a result, it would be possible to increase the efficiency of regional or local air quality management strategies.

### Limitations and Future Work

The current study contains several limitations. Only some of the deep and subsurface underground lines were monitored and only for a small sampling period. In this study, we assumed that these measurements also represented the corresponding lines that were not measured. Moreover, only 23 monitoring stations were available for PM_2.5_ and their locations were not uniformly spread across the study area. Consequently, this may have affected the simulation accuracy and the interpolated ambient concentration estimates in the unmonitored areas that were far from the stations that might have contained higher uncertainty. Furthermore, another limitation was the assumption that the Indoor microenvironment and the average dwelling I/O ratios also represented the office and commercial buildings. The toxicity of PM_2.5_ was not included, but mainly because it was out of the scope of the study to investigate the toxicity of the particles.

The space–time–activity data is based upon the London Travel Demand Survey for the period 2005–2010 and may not be fully accurate locally, spatially and temporally, for the year 2017. Moreover, the annual average of the time-activity data that we used, assuming that people followed the typical daily mobility patterns for the whole year, may have increased the uncertainty in our models, because those data might not have accurately represented a part of the population. Since the main body of our study was based on averages and the population was not divided into different age groups, our health burden predictions may be less accurate for special groups of people that have different behaviours (e.g., ill or elderly that spend most of the day inside their residence).

Parameters that could affect particle infiltration, such as differences in indoor-outdoor air pressure due to the impact of the surrounding micro-environment, and the existence and efficiency of mechanical filtration, were not the focus of the current study and were therefore not investigated.

In the future, this study could be improved by conducting further measurements in the London Underground and for larger periods of time. Simple sensitivity tests could be made in order to check each model’s response and how the misclassification affects our estimates. As the next stage of this work we could investigate how this framework applies to other cities with higher ambient PM_2.5_ concentrations and different indoor characteristics (such as interventions-PACs, HVAC). Taking into consideration that each urban area may have different characteristics, it is important to examine how the incorporation of the local urban or building features could make an impact on exposure concentration estimates and health burden predictions.

## 5. Conclusions

The use of ambient centralized monitoring concentrations as a surrogate for people’s exposure may not provide an accurate representation in a population study. In this study we developed a static exposure approach, commonly used in epidemiology, as our baseline metric and by incrementally enhancing the metric we were able to report the potential impact that the application of different metrics would have on a health outcomes assessment. We demonstrated that studies focusing on centralized monitoring ambient concentrations may show reduced ability to detect the true associations between exposure to PM_2.5_ and health effects due to inadequate spatial variability of the concentration and the absence of people’s mobility. The magnitude of the misclassification related to the inclusion of indoor MEs and the metric’s complexity was large relative to the dynamic nature of human exposure to air pollution.

This analysis illustrates the significance of allowing for population activity and indoor infiltration. The indoor ME showed the highest contribution to the total population exposure (i.e., 83%), while the LU contributed approximately 15%, although people spend only 0.4% of their time there. Consequently, all our models showed lower total exposures than the traditional exposure approach that assumes that the PM_2.5_ concentrations outside the residence are representative of the total population exposure. Particularly, our most complex and accurate Tier-model estimated an approximately 34% lower mean exposure concentration compared with using simply an outdoor concentration.

The exposure misclassification due to home infiltration and underground ME is likely important in assessing the health burden in an urban area because people in cities spend the majority of their time inside the residence or workplace and the pollution concentrations that occur underground are remarkably high. The misclassification between the traditional exposure approach to estimate health outcomes and our most dynamic metric was found to be 1174 mean predicted mortalities in GLA, with the highest numbers observed in Inner London. 

Overall, by quantifying the health burden misclassification we managed to pinpoint the importance of developing a metric that can adequately represent the study population concerned and showed that the use of more dynamic data in epidemiology could significantly increase the accuracy of health impact assessments.

## Figures and Tables

**Figure 1 ijerph-17-01099-f001:**
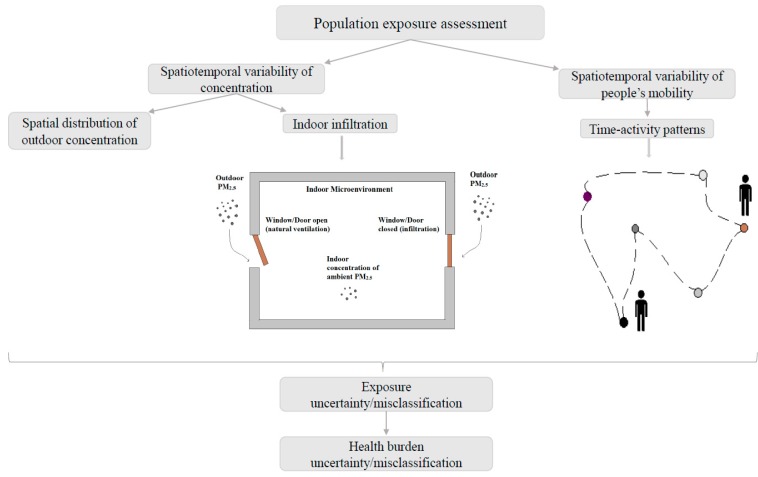
Schematic diagram of an exposure assessment structure for health burden misclassification.

**Figure 2 ijerph-17-01099-f002:**
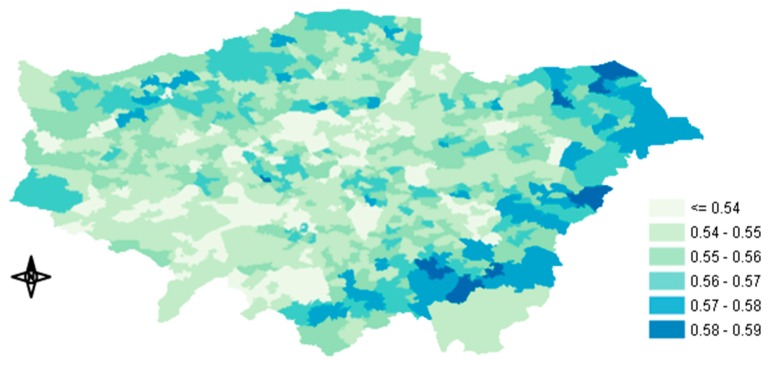
Map of annual average Indoor/Outdoor (I/O) ratios used in our study.

**Figure 3 ijerph-17-01099-f003:**
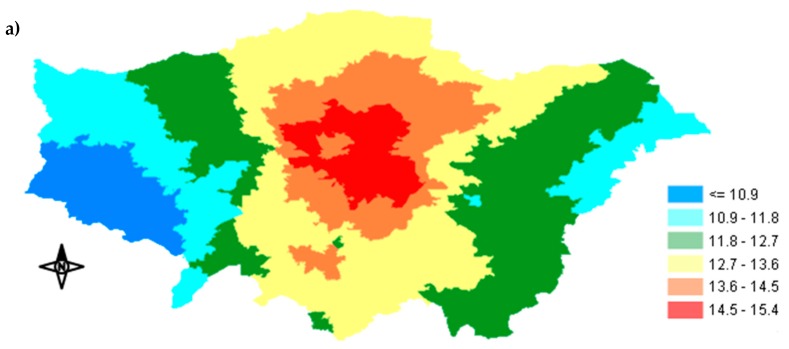
Maps of annual distributions across the Greater London Area (GLA): (**a**) Tier-model 1 annual mean PM_2.5_ exposure concentration (μg/m^3^), (**b**) Tier-model 5 annual mean PM_2.5_ exposure concentration (μg/m^3^), and (**c**) percentage of the PM_2.5_ exposure concentration difference between Tier-model 1 and Tier-model 5.

**Figure 4 ijerph-17-01099-f004:**
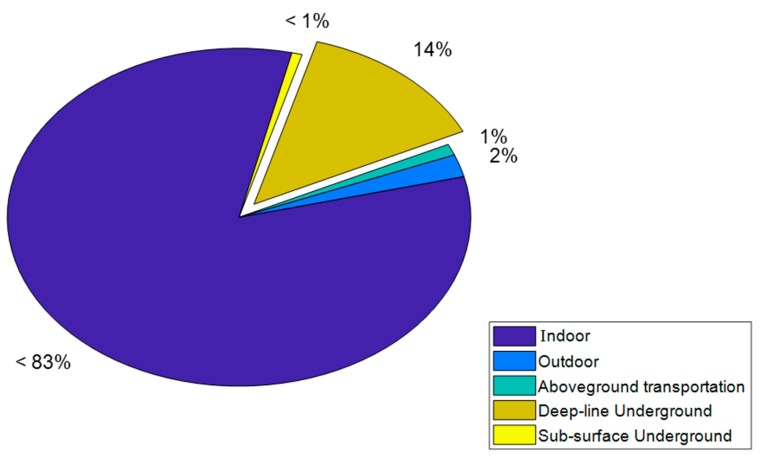
Contribution of each microenvironment (ME) to the total exposure. Indoor exposure shows the greater contribution followed by the deep underground lines.

**Figure 5 ijerph-17-01099-f005:**
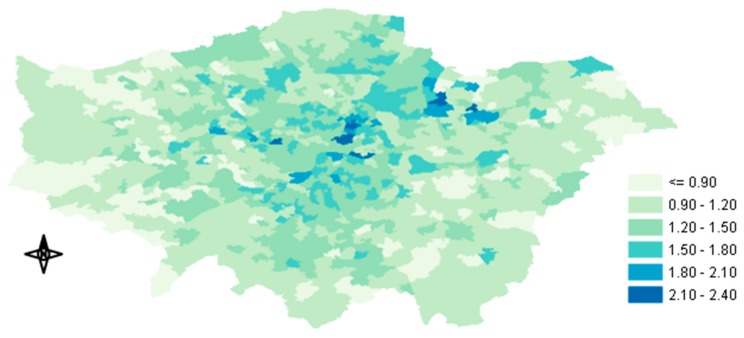
Spatial distribution of the predicted change in the estimates of mortality burden misclassification (in death cases) between Tier model 1 and Tier model 5.

**Table 1 ijerph-17-01099-t001:** Tier models for assessing the time-weighted exposure.

Tier Models	Exposure Equation	Approach
Tier model 1	E = C_out_	Outdoors only
Tier model 2	E = C_ind_	Indoor only
Tier model 3	E = ∑C_out_*F_i_ *x_i_,	Indoor only (dwellings)
Tier model 4	E = (C_out_*t_out_) +(∑C_out_*F_i_ *x_i_)*t_ind_ + (∑C_out_* F_j_)* t_abg_ + (C_undg_*t_undg_)	Outdoor + Indoor + Transportation (abg. and undg.)
Tier model 5	E = (C_out_* t_out_) + [(∑C_out_*F_i_ *x_i_)*t_ind_] + (∑C_out_* F_j_)* t_abg_ + (C_undg-hvac_*t_undg-hvac_)+(C_deep-undg_ *t_deep-undg_)	Outdoor + Indoor + Transportation (abg., deep-line + subsurface undg)

**Table 2 ijerph-17-01099-t002:** London’s dwelling group type descriptions, frequency in stock and average Indoor/Outdoor (I/O) ratios.

Dwelling Type	Frequency %	I/O Ratios	Total Average I/O Ratio (All Dwellings)
Bungalow	1.81	0.63	0.56
Flat	50.4	0.54	
Terraced	28.1	0.56	
Semi-detached	14.5	0.585	
Detached	4.06	0.585	
Unknown	1.13	0.56	

**Table 3 ijerph-17-01099-t003:** Summary table of the time–activity data.

Microenvironments (Groups)	Mode/Place	Time Spent (%)
Outdoor	Walking	1.3
	Cycling	0.1
Transportation (public/private)	Bus	0.7
Indoor	Car	1.6
	Rail	0.2
	Underground/DLR	0.4
	Home, office, other indoor	95.7

**Table 4 ijerph-17-01099-t004:** Annual exposure calculated in each model stage.

Tier Models	Annual Exposure (μg/m^3^)	Standard Deviation (+/– μg/m^3^)
Tier model 1	13.07	1.2
Tier model 2	7.18	0.66
Tier model 3	7.26	0.66
Tier model 4	8.3	0.67
Tier model 5	8.6	0.67

**Table 5 ijerph-17-01099-t005:** Change in the annual mean estimates of mortality (predicted avoided mortalities) between the different exposure metrics and decrease between the estimated change in mortality predictions.

Tier Models	2.5^th^ percentile	97.5^th^ percentile	Mean	Decrease (%)
Tier models 1–2	427	2633	1541	
Tier models 1–3	421	2598	1521	1.95
Tier models 1–4	347	2151	1257	18.4
Tier models 1–5	324	2010	1174	23.8

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
