# Peer review of "Quantifying the Health Burden Misclassification from the Use of Different PM2.5 Exposure Tier Models: A Case Study of London"

_ijerph, 2020, doi:10.3390/ijerph17031099_

Round 1
Reviewer 1 Report
Thank you very much for allowing me to review the original article entitled "Quantifying the health burden misclassification from the use of different PM2.5 exposure tier models: A case study of London" (ijerph-692784).
The starting hypothesis is quantify the misclassification that occurs when using different 11 exposure approaches to predict the mortality burden of a population using London as a case study.
The aim of this work is to move one step further and quantify the misclassification that occurs when using different exposure approaches to predict health burdens. Hence, it develop a framework for quantifying the misclassification of the total mortality burden attributable to exposure to fine particulate matter (PM2.5) in four major microenvironments (MEs) (dwellings, aboveground transportation, London Underground (LU) and outdoors) in the Greater London Area (GLA), in 2017.
They used five different exposure Tier-models.
This is a study that allows quantify the misclassification and give an insight into how the absence of significant MEs from an exposure assessment could increase the uncertainty, they mainly focus on the different infiltration factors of home types and the London Underground. Further understanding as to the impact of an exposure assessment on the mortality predictions and helps to mitigate the uncertainty in health risk assessments of air pollution and as a result, it is possible to increase the efficiency of state or local air quality management strategies.
Comments:
The final part of the introduction indicating the basis of the study design should be incorporated into Material and methods.
Indicate the source of "Time spent" information in table 1.
Indicate in material and methods the design used. I believe that an ecological design has been used on which the five different models have been applied, please indicate the sources of information more specifically (for example table 3).
Has exposure time been considered costly? Or has the cumulative effect of PM2.5 been assessed in any way?
Please check table 4.
Reviewer 2 Report
The manuscript developed a framework coupled with five different exposure Tier-models to quantify the misclassification of the total mortality burden attributable to exposure to atmospheric PM2.5 in four major microenvironments (MEs) in the Greater London Area (GLA). The topic is interesting, but the novelty of the paper is insufficient in its present form. Please address the following comments:
The main results should be concluded in the abstract. The literature review is not up-to-date. An updated and complete literature review should be conducted. The authors should highlight the question: 1) why your proposal is important; and 2) what is the original contribution of the work? There are some minor mistakes; I only listed some of them. Lines 5, 34, 211, 214… Outdoor PM2.5 was extracted from LAQN, the authors should added the some description on how these data are obtained? Are these from the monitored stations? And how the stations are fixed? The authors used the I/O ratio to predict the indoor PM2.5 by considering the dwelling types. While the microenvironment was influenced by surrounding environment, architectural layout, SVF, natural ventilation or vegetation. How do the authors exclude these influences? What is the meaning of Table 4? Please explain. More quantified results should be added in conclusion for more conducive for local administrations, designer and policy makers.Author Response
Please see the attachment.
